# Prenatal Course and Sonographic Features of Congenital Mesoblastic Nephroma

**DOI:** 10.3390/diagnostics12081951

**Published:** 2022-08-12

**Authors:** Theera Tongsong, Watchareepohn Palangmonthip, Wisit Chankhunaphas, Suchaya Luewan

**Affiliations:** 1Department of Obstetrics and Gynecology, Faculty of Medicine, Chiang Mai University, Chiang Mai 50200, Thailand; 2Department of Pathology, Faculty of Medicine, Chiang Mai University, Chiang Mai 50200, Thailand

**Keywords:** congenital mesoblastic nephroma, prenatal course, ultrasound

## Abstract

***Background:*** Congenital mesoblastic nephroma (CMN) is the most common renal tumor among fetuses and infants before the age of 6 months. It usually behaves as a benign tumor. The prenatal features and outcomes of pregnancies with fetal CMN have never been systematically reviewed and analyzed, whereas neonatal or pediatric series have been published several times. The aims of this study are to (1) describe the prenatal natural course and prenatal sonographic char-acteristics of CMN; (2) determine the outcomes of pregnancies with fetal CMN; and (3) demonstrate typical sonographic images together with video clips of prenatal CMN, as an educational example based on our index case presented here. ***Methods:*** Studies focused on fetal CMN, including those consecutively published on PubMed from 1980 to June 2022 as well as the index case presented here, were identified and validated to perform a systematic review. The data of fetal imaging and the prenatal course of pregnancies were extracted for analysis. ***Results:*** The findings derived from 41 cases of review are as follows: (1) No single case has been diagnosed in the first half of pregnancy. No cases were detected during routine anomaly screening at mid-pregnancy. All cases were de-tected in the third trimester or late second trimester. (2) Polyhydramnios is very common and is the first clinical manifestation in most cases, leading to detailed ultrasound in the second half of pregnancy. (3) Preterm birth and low birth weight are the most common adverse pregnancy out-comes, resulting in neonatal morbidity. (4) Hydrops fetalis, though relatively rare, can be associated with CMN and is a grave sign. (5) Prenatal diagnosis is essential since it is critical for the antenatal plan, comprising either referral to a tertiary care center or proper surveillance to prevent serious obstetric complications, especially preterm birth. (6) Ultrasound is the primary tool for prenatal diagnosis of CMN, whereas MRI can be used as an adjunct if some other tumors are suspicious or sonographic features are not typical for CMN. ***Conclusion:*** In contrast to CMN in neonates, fetal CMN is much more serious since it significantly impacts adverse pregnancy outcomes and perinatal morbidity and mortality. The typical prenatal course and the sonographic features of CMN are described.

## 1. Introduction

Congenital renal tumors are relatively rare, including the most common types, with a decreasing order of the prevalence as follows: congenital mesoblastic nephroma (CMN), nephroblastoma (Wilms tumor), rhabdoid tumor, clear cell sarcoma, hamartomas, and ossifying tumor of infancy [1]. CMN is a rare pediatric tumor of the kidney with the highest peak of incidence during the first 3 postnatal months, accounting for 3–10% of all pediatric renal neoplasms. It is the most common renal tumor before the age of 6 months, accounting for nearly half of the renal tumors in that age group, while it constitutes only 5% of renal tumors before 15 years. It usually behaves as a benign tumor and is best treated by surgical resection, which results in cure in most cases without adjuvant therapy [1,2,3,4]. However, although CMN almost always has a favorable prognosis, serious complications can be associated with this tumor, such as preterm labor and birth, polyhydramnios, hydrops fetalis, neonatal hypertension, respiratory distress syndrome, the development of metastases, and hemodynamic failures secondary to a massive space-occupying renal mass, which reduces the survival rate. Accordingly, prenatal diagnosis of this tumor is essential for proper antenatal management, which significantly contributes to better pregnancy outcomes. CMN can be prenatally diagnosed by ultrasound and MRI. Several case/series reports of the prenatal diagnosis of CMN have been scatteringly published in the literature, and they are increasing in number [2,5,6,7,8,9,10,11,12,13,14,15,16,17,18,19,20,21,22,23,24,25,26,27,28,29,30,31,32,33,34,35,36,37,38,39,40]. However, to the best of our knowledge, the prenatal features and outcomes of pregnancies with fetal CMN have never been systematically reviewed and analyzed, whereas neonatal and pediatric series have been published several times. The aims of this study are to (1) describe the prenatal natural course and prenatal sonographic characteristics of CMN; (2) determine the outcomes of pregnancies with fetal CMN; and (3) demonstrate typical sonographic images together with video clips of prenatal CMN, as an educational example based on our index case presented here.

## 2. Methods

### Literature Review

Article selection and data extraction: A comprehensive literature review focused on CMN that was prenatally diagnosed was conducted. The review involved studies published on PubMed from 1980 to June 2022. The following search terms were used: (prenatal [ti] OR fetal [ti]) OR fetus [ti] AND mesoblastic nephroma [ti]. The non-English articles were also included, although some important details could not be extracted. Of all the consecutively searched results, the abstracts and titles were firstly scanned to determine if they matched the selection criteria. The full-text papers of all the selected articles were identified and retrieved for a comprehensive review to extract the data of fetal imaging and the prenatal course of pregnancies. The extracted data from each article were entered into a predefined database form, as presented in Table 1, focusing on the demographic data of the pregnant women, the sonographic features, and obstetric outcomes. The extracted data were descriptively analyzed and reported in percentages and absolute values.

*Statistical analysis:* All statistical procedures were performed using the statistical package for the social sciences (SPSS) software, version 26.0 (IBM Corp. Released 2019. IBM SPSS Statistics for Windows, Version 26.0, Armonk, NY, USA: IBM Corp). The baseline and clinical characteristics were presented as means ± SD for continuous data and percentages for categorical data.

## 3. Index Case with Typical Images as an Educational Tool

A 23-year-old pregnant woman, G2 P1001, was referred to our hospital at 30 + 2 weeks of gestation for detailed ultrasound because of an abnormal mass in the fetal abdomen, suspected to be a gastrointestinal tumor. Her first pregnancy was uneventful, and she gave birth to a term healthy baby weighing 3600 g. The first child was healthy and had normal development. Her current pregnancy course prior to this visit was unremarkable. All basic laboratory tests of antenatal care were within normal limits. At her first antenatal visit, at 20 weeks of gestation, the ultrasound examination performed for fetal anomaly screening revealed no structural abnormalities and normal fetal growth. At 26 weeks of gestation, slightly large-for-date uterine height was suspected by clinical examination. Ultrasound examination showed appropriate fetal size and growth rate. Polyhydramnios with an amniotic fluid index (AFI) of 21 was demonstrated, but no structural abnormality was noted. The follow-up ultrasound scans at 30 weeks of gestation showed progressive polyhydramnios with an AFI of 26 and a solid mass in the fetal abdomen, suspected to be a gastrointestinal tumor. Fetal ultrasound examination at our center demonstrated a live male fetus with appropriate growth, a large left renal tumor, and polyhydramnios with an AFI of 27. The tumor had the following characteristics (Figure 1): located at the left retroperitoneal space of the renal fossa, measured 5.8 × 5.1 × 3.6 cm in diameter, adhered to the left renal parenchyma, solid, rather homogeneous echogenicity similar to that of the renal parenchyma with some areas of low-level echoes, and well-circumscribed and well-demarcated with an echogenic outline, as presented in Figure 1A. The midline structures, including the great vessels, were displaced to the right side. Color Doppler ultrasound demonstrated strong disorganized vascularization, mainly arising from the left renal artery, as presented in Figure 1C, and also supported by 3D ultrasound as presented in Figure 1D. The fetus was in an unstable lie because of polyhydramnios. All other fetal structures, including the left kidney and left adrenal gland, were otherwise normal. No hydropic signs were noted, but hydrocele was clearly demonstrated and probably an early sign of hydrops fetalis. Cardiac function was normal with a myocardial performance index of 0.52 and 0.51 for the left and right sides, respectively. Based on ultrasound findings, the provisional diagnosis was CMN, with differential diagnoses of nephroblastoma (Wilms tumor), rhabdoid tumor, and hamartoma of the kidney. Fetal MRI was not performed since little additional information and no change in management could be expected. A multidisciplinary conference with newborn specialists and pediatric surgeons was held. The plan of management included follow-up ultrasound within 2 weeks to monitor the tumor progression and hydropic signs as well as progressive changes or complications associated with polyhydramnios, fetal surveillance of well-being, and postnatal ultrasound and MRI. At 30 weeks of gestation, the patient had preterm labor. Successful inhibition of preterm labor with nifedipine was achieved, and dexamethasone for promoting lung maturity was given. One week later, at 31 weeks of gestation, the patient developed abdominal discomfort due to polyhydramnios and preterm labor occurred. Amnioreduction (2000 mL) was successfully performed. However, labor inhibition failed. External fetal monitoring showed a reassuring fetal status (category 1). Because the fetus was in an unstable (oblique) lie, an emergency cesarean section was performed, leading to the birth of a male newborn weighing 1685 g, with Apgar scores of 8 and 10 at 1 and 5 min, respectively. Neonatal ultrasound and CT scan confirmed the prenatal findings. The baby underwent a left nephrectomy. The findings revealed an enlarged kidney (tumor) with an intact capsule, which measured 10.3 × 7.0 × 7.0 cm. The pathological findings were as follows: Gross pathological findings (Figure 2A) revealed a 320 g, 10.3 × 7.0 × 7.0 cm left kidney with an attached left ureter and left adrenal gland. The renal capsule was grossly intact. The kidney was bivalved to reveal a large solid mass involving almost the entire kidney. Microscopic pathological examination (Figure 2B–G) revealed characteristic findings of congenital mesoblastic nephroma with mixed classic and cellular types. The area of cellular type was corresponding to the gross hemorrhagic area. Tumor necrosis was present. The tumor involved renal sinus soft tissue, perirenal fat, and the pelvicalyceal system. It focally involved part of the adrenal capsule without adrenal parenchymal invasion. All resection margins were free. One hilar lymph node was negative.

***Comments on the case:*** (1) The images are informative and typical of CMN in terms of a well-demarcated and well-circumscribed solid mass with relatively homogeneous echogenicity and moderate to strong vascularization. (2) It is possible that the tumor had already developed and, with very careful inspection, could have been visualized at 26 weeks of gestation when polyhydramnios had developed. (3) Because of its late occurrence, routine anomaly screening at mid-pregnancy could not detect this type of tumor in this case, as also reported in all previous reports. (4) Large-for-date uterine size or unexplained polyhydramnios in the second half of pregnancy should warrant the possibility of CMN and very careful detailed ultrasound examination of the fetal kidneys must be performed.

## 4. Results

A total of 37 publications were identified. Seven were excluded from the analysis because no data were available [17,20,29,31,34,35,36]. The remaining 30 reports, including 40 fetuses with fetal CMN, were comprehensively reviewed. A total of 41 cases, including our index case, were available for analysis, as presented in Table 1. The mean maternal age was 29.4 years. Most were parous pregnancies (60.9%). Nearly all were naturally conceived, while two of them (4.9%) were conceived by IVF using the transfer of a cryopreserved embryo [28,41].

The prenatal sonographic features of CMN are presented in Table 2. None of them were detected during routine anomaly screening in the second trimester. Most cases were detected in late pregnancy, with a mean gestational age of 32 weeks. The earliest case was diagnosed at 22 weeks of gestation. The main characteristics of the tumors are presented: solid (88%), heterogeneous–homogeneous echogenicity (83.3%), well-demarcated (95.5%), high vascularization (90.9%), and polyhydramnios (65.8%). The tumors occurred predominantly on the left kidneys (64.7%) and showed a preference for male fetuses (67.6%). The mass tended to have a rapid growth, with an average size of 5.2 cm diameter at the time of diagnosis.

Polyhydramnios was the first clinical manifestation in most cases, leading to detailed ultrasound in the second half of pregnancy. Hydrops fetalis, defined as fluid collection in at least two body spaces of the fetus, was noted in three cases [2,41,42]. Two of them were associated with polyhydramnios. All of the three cases were lethal; one experienced intrauterine death and the others died shortly after birth. The most common adverse obstetric outcome was preterm birth (61.5%), with a mean gestational age of 34.7 weeks and a mean birthweight of 2195 g, as presented in Table 3. 

Perinatal death was found in 12.2% of the cases and was associated with hydrops fetalis and immaturity. However, CMN among the fetuses without serious obstetric complications had a favorable prognosis. All the neonates with live birth who underwent surgical management survived and were healthy without adjuvant therapy. Ultrasound is the primary tool for prenatal diagnosis of CMN, whereas MRI was performed prenatally in 26.8% (11/41 cases) and all confirmed the ultrasound findings. 

## 5. Discussion

**New insights** gained from this study are as follows: (1) Although CMN in neonates has a natural course of a very good prognosis after surgical removal, it is much more serious in intrauterine life, since fetal CMN significantly impacts adverse pregnancy outcomes and perinatal morbidity and mortality. (2) No single case has been diagnosed in the first half of pregnancy. No cases were detected during routine anomaly screening at mid-pregnancy. All cases were detected in the third trimester or late second trimester. (3) Polyhydramnios is very common and is the first clinical manifestation in most cases, leading to detailed ultrasound in the second half of pregnancy. (4) Preterm birth and low birth weight are the most common adverse pregnancy outcomes, resulting in neonatal morbidity. (5) Hydrops fetalis, though relatively rare, can be associated with CMN and is a grave sign. (6) Prenatal diagnosis is essential since it is critical for the antenatal plan, comprising either referral to a tertiary care center or proper surveillance to prevent serious obstetric complications, especially preterm birth. (7) Ultrasound is the primary tool for prenatal diagnosis of CMN, whereas MRI should not be routinely performed prenatally.

**Natural course of fetal CMN:** In this review, nearly all the cases of CMN in the prenatal series were detected in the third trimester, although a minority of cases were diagnosed in the late second trimester. None were diagnosed in the first half of pregnancy. It is noteworthy that in most cases, including our index case, the tumor mass size was relatively large at the time of diagnosis. It is possible that a small tumor has similar echogenicity to that of the normal renal pyramid, making it difficult to differentiate it from normal kidneys. Some fetuses, including our case, showed polyhydramnios without a renal mass a few weeks before the diagnosis, implying that detection of the tumor was missed in the early stage. Nevertheless, it is reasonable to conclude that CMN preferentially occurs in late pregnancy, with a rapid increase in tumor size.

It is noteworthy that most cases were conceived naturally. However, two out of the 41 pregnancies (4.9%) of this prenatal series were conceived by IVF, using the transfer of a cryopreserved embryo [28,41]. Additionally, Yiğiter et al. [43] reported a case of CMN in a 2-month-old female infant conceived by IVF and born at 30 weeks of gestation. Accordingly, in view of the low incidence of CMN among natural pregnancies, the repeated occurrence of CMN in IVF pregnancies may suggest a potential causative association. However, such a relationship must be elucidated by future large cohorts.

**Effects on pregnancy outcomes:** (1) Polyhydramnios is identified in about 66% of cases. The pathogenesis of polyhydramnios is unclear. Several mechanisms have been proposed, including the following: (a) Polyuria caused by an increase in renal perfusion and hypercalcemia induced polyuria of the fetus induced by prostaglandins secreted by the renal tumor [44,45]. (b) Bowel obstruction caused by the pressure effect of the renal mass. Polyhydramnios tends to be severe in cases of CMN and some cases need amnioreduction to relieve maternal discomfort. (2) Preterm birth, likely due to polyhydramnios, is the main adverse obstetric outcome associated with neonatal morbidity. Nearly half of the cases (49.5%) in this review had preterm birth (before 37 weeks of gestation). Certainly, preterm birth is strongly associated with several adverse outcomes of the newborns, either respiratory distress syndrome, necrotizing enterocolitis, intraventricular hemorrhage, or long hospital stays in NICU. (3) The cesarean section rate is markedly high in this review, accounting for more than 70% of births. This might be indicated by fetal malposition, likely related to polyhydramnios, and other obstetric complications such as fetal distress or cord prolapse after the rupture of membranes. (4) Hydrops fetalis: CMN can be associated with nonimmune hydrops fetalis, although its mechanisms remain unclear. We postulate that hydrops fetalis may be explained by the following reasons. (a) CMN may be highly vascularized and associated with high output cardiac failure, leading to hydrops fetalis. (b) A large CMN may have pressure effects on the liver, causing obstruction of the portocaval circulation or the major infradiaphragmatic vessels, especially the inferior vena cava and the ductus venosus. All of the three cases in this review ended up with fetal or neonatal death. Accordingly, hydrops fetalis associated with CMN should be considered as an ominous sign.

**Sonographic features:** Prenatal sonographic features are as follows: (1) tumor characteristics: solid, well-demarcated, heterogeneous echogenicity in most cases, homogeneous mixed with some relatively low-level echoes in some cases, and similarity with the echogenicity of the normal renal parenchyma; (2) unilateral; (3) relatively large size at the time of diagnosis, often with displacement of the great vessels to the contralateral side; (4) typically appearing in the late second trimester or third trimester; (5) polyhydramnios in most cases; (6) hydrops fetalis in rare cases, representing 7.3% of cases and considered as a grave sign; (7) high vascularization (moderate to strong). Color Doppler flow can be used to highlight a vascular ring running along the tumor border. Additionally, three- and four-dimensional ultrasound can be used as an adjunct to demonstrate the origin of the tumor, to confirm its renal nativity [27].

Note that polyhydramnios is the first clue, in most cases, leading to prenatal detection of the tumor. Nevertheless, polyhydramnios, commonly found together with a large tumor, is theoretically not a sign of the tumor in the early stage. It is possible that a small tumor which develops early in the second half of pregnancy is subtle and asymptomatic, and not associated with polyhydramnios, resulting in no clinical clues for ultrasound examination.

Careful inspection of the renal parenchyma echogenicity may be helpful in differentiating the tumor mass from normal kidneys. In the early stage, it may be difficult to outline the tumor on prenatal ultrasound because it is contiguous with the normal parenchyma, does not have a well-demarcated capsule as seen in a large mass, and blends with the remaining normal kidney.

In cases of abnormally enlarged kidneys or asymmetrical shape, much more attention must be paid to visualize the normal renal pyramids, which align with the typical renal vessel tree. In early developing tumor, the loss of typical pyramid features and disorganized vascularization on color Doppler flow may be helpful to facilitate early diagnosis or close ultrasound follow-up. Since CMN tends to have a rapid growth, the typical sonographic features are likely to be more obvious on follow-up scans. Additionally, both the adrenal glands and contralateral kidney should be meticulously visualized. If highly suspected, MRI can be helpful to evaluate the origin and morphological characteristics of a fetal abdominal mass [8,10,19].

**Differential diagnosis:** On prenatal ultrasound, fetal renal tumors typically appear as a solid mass in the renal fossa or paraspinal region. The main differential diagnoses of a solid renal mass are as follows: (1) Congenital mesoblastic nephroma (CMN): the prenatal ultrasound characteristics are as mentioned above. (2) Wilms tumor (nephroblastoma) [46,47,48,49]: the tumor appears as a solid echogenic mass with a well-defined capsule with areas of hemorrhage and necrosis, resulting in heterogeneous echodensity. Color flow mapping and spectral Doppler show increased vascularization in the mass with low resistance. In contrast to CMN, bilateral lesions are seen in 5–10% of cases. This tumor is the most common renal childhood malignancy, but it is extremely rare in fetuses, representing 0.16% of all Wilms tumors, with very few studies demonstrating histologic confirmation of Wilms tumors antenatally [47]. It is difficult to differentiate from CMN prenatally. (3) Adrenal neuroblastoma [15,50,51,52]: this is the most common intra-abdominal tumor in neonates. The mass has mixed echogenicity with solid and cystic components. On careful examination, it can be demonstrated as a retroperitoneal mass, separated from the kidney. Elevated levels of amniotic fluid catecholamines support the diagnosis since these tumors usually produce and secrete catecholamines. (4) Adrenal hemorrhage [53,54,55,56]: the mass has heterogeneous echoes with a poorly-defined border. Color flow mapping shows no flow in the mass. The normal kidney should be visualized. Evolution of the mass over time is a clue to the diagnosis. (5) Retroperitoneal teratoma [57]: this is a rare tumor with solid–cystic components but is mainly cystic in appearance. Color flow mapping shows no flow in the mass. The normal kidney should be visualized.

**Clinical impact:** This review provides important information to help physicians taking care of women when counseling couples in cases undiagnosed in prenatal periods, in spite of huge renal masses. This is due to the fact that CMN is a late occurring disorder, whose diagnosis is simply missed at the time of anomaly screening at mid-pregnancy (18–22 weeks of gestation). Accordingly, it should be emphasized that although fetal anatomical survey for anomaly screening is usually performed once at mid-pregnancy, if ultrasound examination is indicated in late pregnancy for any reasons, a re-evaluation of fetal anatomy for late occurring anomalies that may not have appeared on the prior scans should be performed.

Prenatal diagnosis can impact on the plan of management. Couples should be counseled on early detection of preterm labor, which may result in early labor inhibition as well as steroid administration, leading to higher success than inhibition in advanced labor. Because of the very high rate of preterm birth associated with CMN, a plan of birth in a center with the availability of a neonatal intensive care unit should be strongly considered. Early detection also allows parents to prepare for a child who will need more than standard postnatal care, probably resulting in better care management and survival.

Detailed prenatal ultrasound is a single essential tool for the diagnosis of CMN. MRI may be useful as a confirmatory investigation, but it is not necessary in most cases, although some authors recommend it in all cases [10]. We noted that prenatal MRI was used to confirm the ultrasound findings without changing the management plan. Therefore, MRI should be preserved for confusing cases or used as a postnatal work-up for the confirmation of the prenatal findings and for delineation of the tumor as a preoperative plan.

**Recommendations:** (1) If ultrasound examination is not performed routinely in late pregnancy whenever indicated, a re-evaluation of the anatomical survey should be conducted, even in cases with normal scans at mid-pregnancy. (2) In all cases with enlarged kidneys, the possibility of CMN must be kept in mind. A high level of precaution must be exercised because of its similar texture to the normal renal parenchyma, especially in the early development of the tumor. (3) Detailed ultrasound must be performed in all cases of polyhydramnios. (4) Pregnancy with fetal CMN should be monitored for rapid growth, amniotic fluid volume, the early detection of preterm labor and preterm birth, and notification of neonatologists as well as pediatric surgeons. (5) Expectant management is recommended in most cases diagnosed prenatally. However, CMN associated with hydrops fetalis should be strongly considered for delivery for postnatal definitive treatment. (6) The perinatal management plan should be focused on: (a) reliable maternal transportation; (b) effective monitoring of fetal well-being; (c) control of polyhydramnios to prevent preterm labor; (d) early detection of preterm labor and early management; and (e) elective newborn surgery at a stable condition.

## Figures and Tables

**Figure 1 diagnostics-12-01951-f001:**
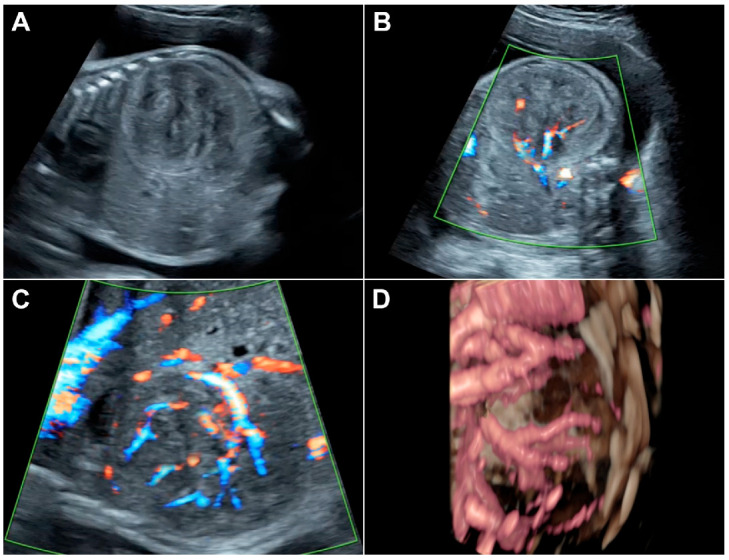
Prenatal ultrasound of the renal mass (congenital mesoblastic nephroma). (**A**) Coronal scan of the fetal trunk shows a well-demarcated, heterogeneous solid mass; (**B**) cross-section of the fetal abdomen shows the same mass; (**C**) color-flow mapping of the mass shows strong vascularization; (**D**) color-flow 3D ultrasound shows strong vascularization.

**Figure 2 diagnostics-12-01951-f002:**
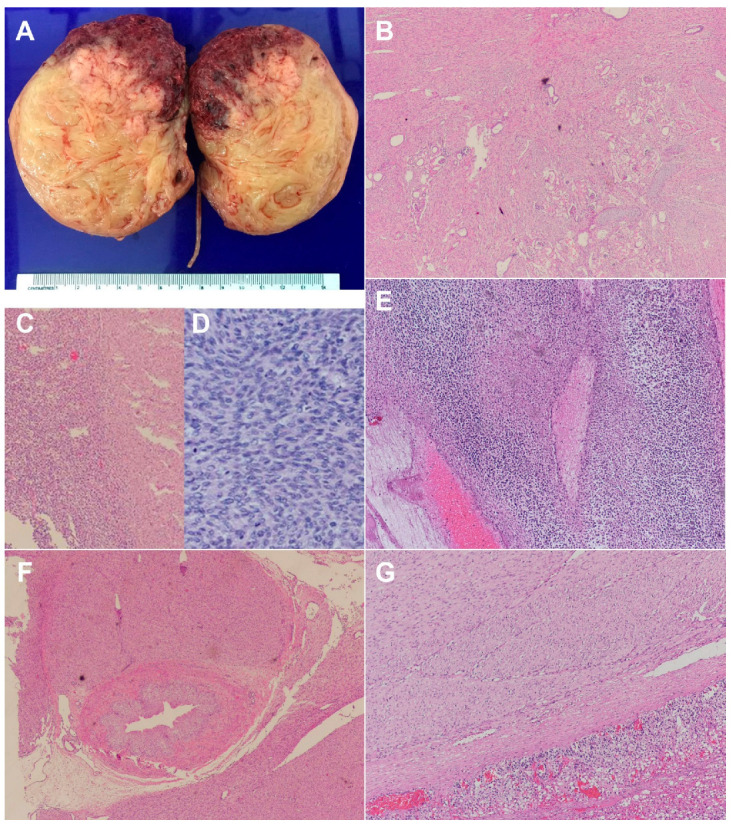
(**A**): The mass showed a white–tan, firm, whorled appearance with a poorly defined border. The superior portion of the mass revealed a softer consistency with hemorrhage. Normal renal parenchyma was seen medially; (**B**): (4× magnification) Classic-type, bland spindle cells arranged in intersecting fascicles with entrapped renal parenchyma and islands of cartilage; (**C**): (10× magnification) Junction between classic type (right portion) and cellular type (left portion); (**D**): (40× magnification) Cellular-type, plump spindle cells with high mitotic activity arranged in sheetlike growth pattern; (**E**): (10× magnification) Tumor necrosis; (**F**): (4× magnification) Tumor involved renal sinus soft tissue close to the ureter; (**G**): (10× magnification) Tumor involved adrenal capsule.

**Table 1 diagnostics-12-01951-t001:** Summary of the clinical and sonographic characteristics of individual case reports of congenital mesoblastic nephroma.

Author	Age	Parity	GA dx	Size	Poly	MRI	Side	Echo	Solid	Border	Vascular	Route	Sex	GA	BW	APG -1	APG -5	Prognosis	FU -Day	Postnatal Treatment
Liu 2022 [5]	36	0	36	5.8	Yes	Yes	Left	Heterogeneous	Solid	Well	Mod	CS	Male	38	3560	9	10	Survive	1200	Nephrectomy
Lin 2021 [6]	31	0	32	5.7	Yes			Homogeneous	Solid	Well	Strong	CS		32	1802	7	9	Survive	21	Nephrectomy
Chen 2021 [7]	26		37	3.5	No		Right					Vg	Male	38				Survive	1740	Nephrectomy
	30		25	2.8	No		Left					CS	Female	39				Survive	360	Nephrectomy
	31		31	3.7	Yes		Left					Vg	Male	35				Survive	360	Nephrectomy
	22		39	3.3	No		Left					Vg	Female	40				Survive	1650	Nephrectomy
	28		38	5.0	No		Right					Vg	Female	39				Survive	360	Nephrectomy
	31		31	5.4	Yes		Right					CS	Male	32				Survive	990	Nephrectomy
	25		37	4.6	No		Right					Vg	Male	38				Survive	540	Nephrectomy
	40		35	6.9	No		Right					CS	Male	35				Survive	480	Nephrectomy
	33		32	4.6	Yes		Left					CS	Male	35				Survive	420	Nephrectomy
	27		37	4.1	No		Left					Vg	Male	39				Survive	390	Nephrectomy
	30		29	3.7	Yes		Left					CS	Male	31				Survive	360	Nephrectomy
Che 2021 [8]	29	0	35	2.9	Yes	Yes	Right	Homogeneous	Solid	Well	Mod	Vg	Female	38	3250	5		Survive	180	Nephrectomy
Mata 2019 [9]	36	1	28	4.7	Yes	Yes	Left	Heterogeneous	Solid	Well	Strong	CS	Female	34	2150	9	10	Survive	180	Nephrectomy
Manganaro 2018 [10]	34	0	32	3.7	Yes	Yes								32	1500			Survive		Nephrectomy
Do 2015 [11]	39	0	35	5.2	Yes		Right	Homogeneous	Solid	Well		Vg	Male	35	2550	9	10	Survive	240	Nephrectomy
Takahashi 2014 [12]	28	0	23	3.3	Yes	Yes	Left				Strong	CS	Female	28	1210	3	6	Survive		Resection
Ko 2013 [13]	30		30	9.0	Yes	Yes	Left	Heterogeneous	Solid–cystic		Strong	CS	Female	30				Survive	210	Resection
Esmer 2012 [14]	28	1	40	5.2	No	No	Left	Heterogeneous	Solid–cystic	Well		Vg	Female	40	2870	8	10	Survive		Nephrectomy
Montaruli 2012 [16]			36			Yes	Right		Solid	Well				40				Survive		Nephrectomy
Kim 2005 [18]	28	1	34	6.9	No	No	Right	Heterogeneous	Cystic	Well		CS	Female		1750	5	9	Dead		None
Yamamoto 2006 [19]	25	1	34	6.8	Yes	Yes	Right	Heterogeneous	Solid	Well		CS	Male	35	2904	8	9	Survive	330	Nephrectomy
Siemer 2004 [22]	32	0	32	5.5	Yes	No	Right	Heterogeneous	Solid	Well		CS	Male	35		9	9	Survive		Nephrectomy
Chen 2003 [2]	20	0	22	5.0	Yes	Yes	Left	Hypoecho	Solid	Well	No	Vg	Male	25	1030	2	2	Dead		None
Fuchs 2003 [23]	39	2	36	4.6	No	No	Left	Homogeneous	Solid	Well	Strong	CS	Male	37	2430	9	10	Survive		Nephrectomy
Goldstein 2002 [24]	30	2	28	5.9	Yes	No	Left	Hypoecho	Solid	Well		CS	Male	30	1420	2	8	Survive	540	Resection
Won 2002 [25]	28	0	35	5.1	No	Yes	Right	Homogeneous	Solid	Well	Mod	CS	Female	38	3305	6	9	Survive		Nephrectomy
Irsutti 2000 [26]	25		35	3.0		Yes	Left	Homogeneous	Solid	Well		CS	Male	38	2730	9	9	Survive	300	Nephrectomy
Schild 2000 [27]	29	0	34	5.6	No	No	Left	Homogeneous	Solid	Well	Strong	CS		38		10	10	Survive	365	Nephrectomy
Shibahara 1999 [28]	37	0	28	5.7	Yes		Left	Heterogeneous	Solid	Well		CS	Male	34	2564	6	7	Survive		Nephrectomy
Haddad 1996 [30]			33		Yes	No								35				Survive	300	Nephrectomy
Sailer 1993 [32]			32					Heterogeneous	Solid			CS		35	2500	8	10	Survive		Resection
Boulot 1989 [33]			33		Yes	No								33				Dead		None
Walter 1985 [37]	27	0	31	6.5	Yes	No	Left	Homogeneous	Solid	Well		CS	Female	34	2030	6	6	Survive		Resection
Howey 1985 [38]	26	1	30	7.1	Yes	No	Left		Solid			CS	Male		1480	4	7	Survive		Resection
Geirsson 1985 [39]	22	0	27		Yes			Hypoecho	Solid	Poorly			Male	30	1500			Survive		Nephrectomy
Ehman 1983 [40]	25	1	35	4.0	Yes	No	Left	Hypoecho	Solid	Well		CS	Male	38	2500	9	9	Survive		Nephrectomy
De Paene 2011 [41]	35		26	8.8	No	No		Heterogeneous		Well		CS	Male	26	970	0	0	Dead		None
Liu 1996 [42]	23	0	28	5.9	Yes	No	Left	Heterogeneous	Solid			CS	Male	34	3000	1	0	Dead		None
Index case	23	1	30	4.8	Yes	No	Left	Heterogeneous	Solid	Well	Strong	CS	Male	31	1685	8	10	Survive	730	Nephrectomy

APG: Apgar scores; BW: birth weight; CS: cesarean section; GA: gestational age; Vg: vaginal delivery.

**Table 2 diagnostics-12-01951-t002:** Prenatal course and sonographic features.

Parameters	Number (Percentage) or Mean ± SD
Solidity:	
Mainly solid	22/25 (88.0%)
Solid–cystic	2/25 (8.0%)
Mainly cystic	1/25 (4.0%)
Echogenicity:	
Heterogeneous	12/24 (50.0%)
Homogeneous	8/24 (33.3%)
Low-level echoes	4/24 (16.7%)
Border outline:	
Well-demarcated	21/22 (95.5%)
Poorly demarcated	1/22 (4.5%)
Vascularization:	
No	1/11 (9.1%)
Moderate	3/11 (27.3%)
Strong	7/11 (63.6%)
Polyhydramnios:	
No	13/38 (34.2%)
Presence	25/38 (65.8%)
Hydrops fetalis:	
No	38/41 (92.7%)
Presence	3/41 (7.3%)
Sidedness:	
Left	22/34 (64.7%)
Right	12/34 (35.3%)
Size (average diameter; cm)	5.2 ± 1.5 (range: 2.8–9.0)
Gestational age at diagnosis (weeks)	32.2 ± 4.2 (range: 22–40)

**Table 3 diagnostics-12-01951-t003:** Summary of pregnancy baseline and outcomes.

Parameters	Number (Percentage) or Mean ± SD
Maternal age (years)	29.4 ± 5.0 (range: 20–40)
Parity:	
Nulliparous	14/23 (60.9%)
Parous	9/23 (39.1%)
Route of delivery:	
Vaginal	10/35 (28.6%)
Cesarean	25/35 (71.4%)
Gestational age at delivery (weeks)	34.7 ± 3.9 (range: 25–40)
Preterm delivery:	
No	15/39 (38.5%)
Yes	24/39 (61.5%)
Birth weight (grams)	2195 ± 756 (range: 970–3560)
Fetal sex:	
Male	23/34 (67.6%)
Female	11/34 (32.4%)
Postnatal treatment:	
None	5/41 (12.2%)
Resection	6/41 (14.6%)
Nephrectomy	30/41 (73.2%)
Survival:	
Survive	36/41 (87.8%)
Dead	5/41 (12.2%)

## Data Availability

The data of this report are available from the corresponding authors upon request.

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
