# Peer review of "Prenatal Course and Sonographic Features of Congenital Mesoblastic Nephroma"

_diagnostics, 2022, doi:10.3390/diagnostics12081951_

Round 1

Reviewer 1 Report

The authors present a review of literature about the perinatal management of congenital mesoblastic nephroma on the basis of an index case. The study is well conducted and the manuscript is well written. However, there remains only one issue to be clarified. The authors should specify the indication for the cesarean delivery of their index case (previous cesarean, malposition or else). Whenever this specification has been made, I think that the manuscript can be accepted for publication in Diagnostics.

Author Response

Response: The indication for cesarean delivery is now specified, as highlighted (in blue) in “Case” section on page 3.

Reviewer 2 Report

To authors,

1.     Case: Please shortly describe the reason why you did not perform MRI.

2.     Case: Please shortly describe why you did not try (attempt) vaginal delivery. If possible, please describe CTG (cardiotocography) findings.

3.     Line 290: You touched that in some occasions MRI might be of use whereas in abstract you state that MRI is not needed. I have vague feeling/understanding for/of your statement; you might wish to say, “in ordinary (in almost all) cases, ultrasound can identify the condition but if some other tumors are suspicious we had better employ MRI”, right? Please very shortly describe this meaning in Abstract. 

Author Response

  1. Case: Please shortly describe the reason why you did not perform MRI.

Response: The reason is added as highlighted in “Case” section page 3.

  1. Case: Please shortly describe why you did not try (attempt) vaginal delivery. If possible, please describe CTG (cardiotocography) findings.

Response: As highlighted in blue in “Case” section page 3, indication for cesarean section is unstable lie. CTG is now described (category 1), as highlihged.

  1. Line 290: You touched that in some occasions MRI might be of use whereas in abstract you state that MRI is not needed. I have vague feeling/understanding for/of your statement; you might wish to say, “in ordinary (in almost all) cases, ultrasound can identify the condition but if some other tumors are suspicious we had better employ MRI”, right? Please very shortly describe this meaning in Abstract.

Response: The statement is now clarified, as highlihged in “Abstract”. Thank you very much for the comment.
